# Altered Sphingolipids Metabolism Damaged Mitochondrial Functions: Lessons Learned From Gaucher and Fabry Diseases

**DOI:** 10.3390/jcm9041116

**Published:** 2020-04-14

**Authors:** Margarita Ivanova

**Affiliations:** Lysosomal and Rare Disorders Research and Treatment Center, Faitfax, VA 22030, USA; mivanova@ldrtc.org

**Keywords:** Gaucher disease, Fabry disease, sphingolipids, mitochondria, autophagy

## Abstract

Sphingolipids represent a class of bioactive lipids that modulate the biophysical properties of biological membranes and play a critical role in cell signal transduction. Multiple studies have demonstrated that sphingolipids control crucial cellular functions such as the cell cycle, senescence, autophagy, apoptosis, cell migration, and inflammation. Sphingolipid metabolism is highly compartmentalized within the subcellular locations. However, the majority of steps of sphingolipids metabolism occur in lysosomes. Altered sphingolipid metabolism with an accumulation of undigested substrates in lysosomes due to lysosomal enzyme deficiency is linked to lysosomal storage disorders (LSD). Trapping of sphingolipids and their metabolites in the lysosomes inhibits lipid recycling, which has a direct effect on the lipid composition of cellular membranes, including the inner mitochondrial membrane. Additionally, lysosomes are not only the house of digestive enzymes, but are also responsible for trafficking organelles, sensing nutrients, and repairing mitochondria. However, lysosomal abnormalities lead to alteration of autophagy and disturb the energy balance and mitochondrial function. In this review, an overview of mitochondrial function in cells with altered sphingolipid metabolism will be discussed focusing on the two most common sphingolipid disorders, Gaucher and Fabry diseases. The review highlights the status of mitochondrial energy metabolism and the regulation of mitochondria–autophagy–lysosome crosstalk.

## 1. Mitophagy

Since the lysosomal and mitochondrial functions are intricately related and critical for maintaining cellular homeostasis, it is intuitive to assume that autophagy-mitophagy processes are affected in Gaucher disease (GD) [1,2]. Usually, mitophagy is initiated by the damaged mitochondrion itself to compensate for the energy crisis [3]. In an effort to prevent cell death, damaged mitochondria trigger mitophagy initiation via the ubiquitination of mitochondrial outer membrane proteins, which are recognized by mitophagy receptors [4]. In the next step, activated mitophagy receptors recruit the microtubule-associated protein 1 light chain 3 (LC3) and GABA type A receptor-associated protein (GABARAP) proteins and trigger the autophagy flux process. During autophagic flux, cytosolic LC3-1 is conjugated to form LC3-II, and LC3-II is incorporated into the autophagosomal membrane. Some data suggest that ceramide (Cer) anchors LC3B-II autophagosomes to mitochondrial membranes to induce mitophagy [5]. Sequestosome 1 **(**SQSTM1/p62) links autophagic cargo to the autophagic membrane [6,7]. The next step is the fusion of autophagic vacuoles with lysosomes to form autophagolysosomes, where the macromolecular components are broken down into metabolites [8]. Autophagosomes maturate into autolysosomes by one of the two pathways: fuse first with late endosomes to form amphisomes, then with lysosomes, or, autophagosomes fuse directly with lysosomes [9]. Then, autophagic cargo is delivered to lysosomes where materials are degraded by acidic lysosomal hydrolases.

## 2. Gaucher Disease and Mitochondrial Function

### 2.1. Gaucher Disease

Gaucher disease (GD) (Online Mendelian Inheritance in Man (OMIM) 23080, 231000, 231005), the most common lysosomal storage disorder (LSD), is caused by mutations in the glucocerebrosidase gene *(GBA)* (OMIM 606463) which results in the deficiency of lysosomal enzyme glucocerebrosidase (GCase) (Enzyme commission number (EC) 3.2.1.45). The *GBA* mutations lead to misfolding of GCase protein in the endoplasmic reticulum, inhibition of protein trafficking to the lysosomes, and, as a result, inhibition of GCase enzymatic activity [10]. The deficiency of GCase results in chronic accumulation of its substrate glucosylceramide (Gl-1) in lysosomes. Recessively inherited GD presents with a broad spectrum of symptoms encompassing primary nervous system, immune system involvement, enlarged spleen and liver, anemia, low thrombocyte counts due to bone marrow involvement, and severe skeletal disorder with pain and permanent disabilities [11,12]. Three distinguished phenotypic presentations of GD are described based on increasing severity. GD type 1 is a non-neuropathic form of GD, GD type 2 and 3 are termed neuronopathic forms of GD.

GCase catalyzes the cleavage of the glycolipid glucosylceramide (Gl-1) into glucose and ceramide (Cer) [13]. The lipid tails of glycolipid, GCase, and the reaction facilitator saposin C (SAPC) are embedded within the intralysosomal membrane where cleavage occurs (Figure 1) [14]. Gl-1, as with glycosphingolipids in general, is involved in a large number of cellular processes, including signal transduction, membrane trafficking, or cytoskeletal processes [13,15]. Since Gl-1 is a primary precursor of complex glycosphingolipids, synthesis and degradation of Gl-1 are crucial steps in sphingolipid metabolism (Figure 1) [16]. In the absence of functional GCase in GD, glucosylceramide is deacetylated to form Lyso-Gl-1 [17,18]. When sphingolipids accumulate in the lysosomes, the pH increases and mediates lysosomal destabilization [19,20,21]. Therefore, dysfunctional lysosomes with excessive levels of Lyso-Gl-1 amass in every cell and interfere with cellular pathways outside the lysosomes.

### 2.2. Gaucher Disease and Mitophagy

Lysosomal dysfunction in LSD is associated with mitochondrial dysregulation and accumulation of damaged mitochondria. What happens with the mitophagy process in GD when lysosomes do not function properly? Increased mitochondrial fragmentation due to inhibition of autophagy was described in midbrain neurons and astrocytes in GBA^−/−^ mice [22]. With inhibition of autophagy flux, decreasing LC3-II level has been established in GD cells, including patients’ macrophages, peripheral blood mononuclear cell (PBMC), and induced pluripotent stem (iPSC) neuronal cells [1,23,24]. Neuronal accumulation of Gl-1 in mouse models of GD leads to accumulation of various autophagic cargo, including dysfunctional mitochondria, ubiquitinated protein aggregates in autophagosomes and lysosomes in the brain, neurons, and astrocytes [22,25,26]. Moreover, accumulation of autophagy flux indicator, SQSTM1/p62, in GD fibroblasts, neurons, or GBA knockout mouse models, provides more evidence of inhibition of the autophagy–lysosomal process [1,23,27].

Recently, emerging studies have demonstrated that the mechanism of inhibition of mitophagy in GD is dual. First, as described before, accumulation of Gl-1 in lysosomes blocks the degradation of autophagic cargo. Second, the accumulation of mutant GBA proteins are responsible for accumulation of unwanted/misfolded proteins in endoplasmic reticulum (ER) [25,26,27,28]. This process leads to ER stress, and evokes the unfolded protein response (UPR) and endoplasmic reticulum-associated degradation (ERAD) [28]. The chronic activation of UPR and ERAD leads to apoptosis. [29]. Moreover, accumulation of unfolded mutated GBA protein leads to inhibition of alpha-synuclein degradation, which directs alpha-synuclein aggregation in GD cells [30,31]. This effect may be attributed to the endoplasmic reticulum (ER) retention due to inability to correctly fold the mutant form of GBA protein and inhibition of proteasomal degradation [32]. The effect of mutant GBA on mitochondrial priming and autophagy induction indicates a gain-of-toxic-function of the mutant protein [33].

### 2.3. Ceramides and Mitochondria Membrane Damage in GD

Is the mitochondrial membrane damaged in GD cells? One of the markers of mitochondrial damage is a change of mitochondrial membrane potential (ѰM). The generation of ѰM by the mitochondrial electron transport chain drives the adenosine di-phosphate (ADP) to adenosine tri- phosphate (ATP). The loss of mitochondrial function, inhibition of oxygen consumption due to decreasing ѰM, has been demonstrated in GD neuroblastoma cells [34], fibroblasts from GD patients (L444P/L444P) [35]. The reason for that is that Cer is a structural component of the cell membrane with an important role in maintaining barrier function and fluidity. Both Cer and Gl-1 define the biophysical properties of the cellular membranes and their functional plasticity [36,37]. Cer and Gl-1 are regulated by the opposite actions of two enzymes: GCase and UDP-glucosylceramide synthase (UGCG) (Figure 1). UGCG forms Gl-1 in the ER by converting Cer (Figure 1). In spite of the significant chemical difference between Cer and Gl-1, they both affect the properties of fluid model membranes but differ in their capacity to promote changes in the cellular membrane shape [38,39]. Gl-1 induces membrane perturbation in a pH-dependent manner. Thus, the analysis of the Cer/Gl-1 ratio suggests decreased membrane fluidity in primary GD cells and cells treated with GBA inhibitors [40]. If instability of Cer/Gl-1 ratio in GD cells may play a role in the formation of number and shape of lysosomes, then it may add a further layer of complexity to the mitochondrial membrane structure too. Moreover, Cer can self-assemble in the mitochondrial outer membrane to form large stable channels capable of releasing apoptotic proteins, for example, allow passage of cytochrome c [41,42]. For example, Cer directly activates apoptosis via the formation of Cer channels in the mitochondrial outer membrane favoring Bcl-2 homologous antagonist/killer-BCL2 associated X protein (BAK/BAX ) activation and regulates caspase 3 by compartmentalization in the late endosomes [43,44].

### 2.4. Ambroxol Therapy for GD Patients and Mitochondrial Function

Enzyme replacement therapy (ERT) is successful for the treatment of type 1 GD with systemic symptoms, including splenomegaly, hepatomegaly, thrombocytopenia, and platelet disorders [45]. However, ERT is not effective for the neurological form of GD due to the weak entry of the recombinant GCase enzyme through the blood-brain barrier (BBB) [46]. Enzyme-enhancement therapy (EET) is based on the ability of small molecules, named molecular chaperones, to fold the misfolded mutant enzyme. This treatment approach has the potential to cross the BBB and to treat the neurological symptoms of GD [47]. One of these small molecules with the prediction of good BBB penetration, ambroxol, was identified as a pharmacological chaperone for GCase [48,49]. Ambroxol therapy also has an excellent safety record, even with high doses [50,51,52]. Ambroxol has unique chaperone characteristics: it works at neutral and acidic pH of the endoplasmic reticulum (ER) and lysosomes. At pH 4.3, near lysosomal pH, ambroxol does not inhibit the enzyme but actually becomes an activator of GCase activity [48,53]. Ambroxol demonstrated the dual independent mechanism of action: increasing GCase activity and activation of the autophagy–lysosomal pathway [53,54,55]. Moreover, a study of the effect of ambroxol on primary cortical neurons suggests that ambroxol increases mitochondrial content via activation of peroxisome proliferator-activated gamma coactivator 1-alpha (PGC1-α) [55]. These emerging findings demonstrate that additional studies are needed to explore the underlying mechanism of activation of mitochondria in the presence of ambroxol. Additionally, ambroxol showed significant enhancing activity of wild-type and mutant (p.A156V and p.R301Q) forms of α-Gal A enzyme, suggesting that ambroxol could potentially be used in the treatment of Fabry diseases [56].

## 3. Fabry Disease and Mitochondrial Function

### 3.1. Fabry Disease Introduction

Fabry disease (FD) is an X-linked disorder, where mutations in the *GLA* gene result in a deficiency of the enzyme α-galactosidase A (α–Gal A) (EC entry 3.2.1.22). α–Gal A catalyzes the lysosomal hydrolysis of globotriaosylceramide (Gb-3) to lactosylceramide and digalactosylceramide (Gal-Gal-Cer) to galactosylceramide (Gal-Cer) [57,58]. The deficiency of this enzyme leads to an accumulation of Gb-3, its metabolite, globotriaosylsphingosine (Lyso-Gb-3), and Gal-Gal-Cer in lysosomes (Figure 1) [59]. Detection of Gb-3, Lyso-Gb-3, and Gal-Gal-Cer in urine and plasma are the standard diagnostic methods for FD [58]. The abnormal buildup of Gb-3 and Lyso-Gb-3 links to cellular dysfunction, that triggers a cascade reaction that causes progressive damage in multiple organs. Hemizygous FD males have a progressive buildup of Gb-3, particularly in the endothelial cells, and Lyso-Gb-3 in the kidneys (smooth muscle cells, and podocytes) and cardiac tissue (including valves, cardiomyocytes, nerves, and coronary arteries) [60,61,62]. Recent metabolic studies showed that Lyso-Gb-3 analogs (Lyso-Gb-3(−28), Lyso-Gb-3(−2), Lyso-Gb-3(+16), Lyso-Gb-3(+18), Lyso-Gb-3(+34), Lyso-Gb-3(+50)) are higher in GLA knockout mice and also present in plasma and urine samples in FD patients with the highest Lyso-Gb-3 levels [63,64,65]. However, the role of Lyso-Gb-3 analogs in FD cellular pathology is unknown.

The signs and symptoms of FD are quite heterogeneous and include not only renal failure or cardiovascular disease, but also cerebrovascular complications, including ischemic or hemorrhagic strokes, dermatologic manifestations, ocular and hearing complications, auditory, and neurologic complications, all of which are associated with reduced quality of life and early mortality [66]. The first clinical symptoms, such as pain in bones and/or joints, occur during childhood or adolescence. Common misdiagnosis for FD is rheumatic fever or somatoform pain disorders. The involvement of the central nervous system in FD increases the incidence of ischemic strokes, a significant cause of the short lifespan in Fabry patients. The life expectancy of male patients with FD, if untreated, is approximately 40–42 years. Heterozygous females have higher residual α–Gal A activities. However, females develop clinical manifestations of varying severity and also have a reduced life span [67]. The subsequent lysosomal dysfunction due to Gb-3 accumulation alters cell signaling pathways [1,24,60]. Since the functional integrity of lysosome–autophagy–mitochondria crosstalk is vital for cell “health,” it is obvious that mitochondrial metabolism will be affected [68]. However, unfortunately, mitochondrial function and energy metabolism were never systematically studied in aspects of FD pathology.

### 3.2. Cardiac Energy Metabolism in Fabry Disease

Cardiovascular pathology is a hallmark of FD. The deposition of Gb-3 and less Lyso-Gb-3 inside the myocardium of FD patients affects the cardiac function, but secondary functional alterations also play a role in progressive cardiovascular pathology [69]. Deposition of Gb-3 in FD cardiomyocytes (CMs) displayed an increased excitability, with altered electrophysiology and calcium handling [70]. The importance of mitochondrial function for the maintenance of energy metabolism for the normal cardiac function is well documented. Machann et al. show that inhibition of mitochondrial function, and thus energy metabolism, play a significant role in FD cardiomyopathy [69]. Magnetic resonance spectroscopy showed a reduction of phosphocreatine (PCr), ADP, AMP, and ATP in the left ventricular mass in FD patients [69]. Electron microscopic evaluation of cardiomyocytes from FD patient′s hearts demonstrated that the percentage area of mitochondria in the cytoplasm was reduced and Gb-3 accumulation was increased in 2 from 3 cases [71]. Clinical studies confirmed that dysfunction of cardiac energy metabolism and increased oxygen requirements via left ventricular (LV) hypertrophy are linked to reducing ischaemic tolerance in FD patients [69,71]. How does a lysosomal enzyme deficiency translate into mitochondrial dysfunction in patients with FD. ? Gb-3 accumulation induced reactive oxygen species (ROS) production, suppressed mitochondrial antioxidant superoxide dismutase 2 (SOD2), and enhanced AMP-activated protein kinase (AMPK) activation in vascular endothelial cells and iPS cells derived from FD patients [72,73]. The cardiomyocytes (CM) differentiated from GLA-null embryonic stem cells (ESCs) showed accelerated level apoptosis due to impairment of protein degradation and autophagic flux in the presence of Gb-3 accumulation [74]. Also, MitoSOXRed staining demonstrated an increased level of **residual oxygen consumption** (ROX)production in GLA-null CMs [74]. Therefore, dysfunctional mitochondria could be a source of ROS elevation. Moreover, CM membrane lipid composition is altered in FD and leads to a direct effect on the mitochondrial inner membrane [75]. The respiratory chain complexes I, II, III, and IV are embedded in the mitochondrial membrane and thus could be affected by membrane lipid composition status [76]. Fabry CMs have a lower ѰM compared with control, which could indicate a functional mitochondrial deficiency [70].

### 3.3. Gb-3 and Metabolic Pathways Heavily Implicate FD Nephropathy

Although end-stage renal disease is one of the leading causes of death in male FD patients, the mechanism of kidney failure is not well understood. Progressive Lyso-Gb-3 and less Gb-3 accumulation lead to histological damage of kidney cells, resulting from lysosome rupture [77]. Histological studies have demonstrated that the buildup of Gb-3 in podocytes plays an essential role in the pathogenesis of glomerular damage [78]. Moreover, the study of urine-derived FD kidney epithelial cells and podocytes in vitro showed that the cause of this damage might lie in deregulated autophagy pathways [78,79]. An increase in the autophagosomal activity in FD kidney epithelial cells and podocytes was linked to the upregulation of LC3-II and the downregulation of mechanistic target of rapamycin kinase (mTOR) kinase activity [79]. Also, a high level of Lyso-Gb-3 plays an essential pro-inflammatory role in cultured podocytes, mainly through activation of the Notch-1 signaling pathway [61]. Furthermore, the upregulation of Notch-1 leads to podocyte injury in vivo and kidney fibrosis. Interestingly, the Notch-1 signaling pathway regulates energy metabolism, including glycolysis, Krebs cycle, oxidative phosphorylation, and glutamine metabolism [80].

Medullary thick ascending limbs (mTALs) plays a critical role in the urine concentrating mechanism. TALs are located outside the vascular bundles with limited oxygen supply, which makes them vulnerable to hypoxic injury [81]. TALs cells in GlatmTg(CAG-A4GALT) mice demonstrated flattened cristae and round mitochondria [82]. In contrast, control cells present extensive cristae with a high number of elongated mitochondria. This result suggests that mitochondrial dysfunction due to Gb-3 accumulation may affect mTAls’ function [82]. Overall, this data indicates that dysregulation of sphingolipid metabolism may complicate mitochondrial function in FD kidney cells. It is important to emphasize that the role of mitochondrial metabolism in FD nephropathy remains unknown.

## 4. mTOR Pathway in Gaucher and Fabry Pathology

Lysosomal abnormalities lead to alteration of autophagy and disturb the energy balance and mitochondrial function. The effect is similar in both pathologies, although the type of substrate accumulating in lysosomes differs in Gaucher and Fabry disorders. One of the critical regulators of both the autophagy–lysosomal fusion process and mitochondrial function is the mTOR-dependent signaling pathways [83]. mTOR is docked on the lysosomal surface in its active state inhibiting autophagy and lysosomal fusion until the accumulation of amino acids in the lysosomal lumen triggers autophagy activation [84,85]. Reduction of mTOR activity has been shown in FD fibroblasts and podocytes [78,86], in GD PBMC, and several models of neuronopathic and non-neuronopathic GD, including the *Drosophila melanogaster* GD model [1,87,88,89,90]. It is apparent that the accumulation of sphingolipid substrates in lysosomes inhibits autophagy–lysosome fusion and disrupts the mTOR activation/inactivation cycle.

mTOR acts as a sensor of nutrients and growth factors in the cells and its activity is controlled by the ATP:ADP balance. In response to decreasing ATP levels, AMP-activated protein kinase (AMPK) inhibits mTOR activity [91]. Next, mTOR promotes dephosphorylates TFEB, and TFEB changes subcellular localization from cytoplasm to nucleus and status of transcription activity to trigger autophagy [92,93]. Moreover, mTOR regulates mitochondrial metabolism and biogenesis by promoting the translation of nuclear-encoded mitochondria-related proteins, including mitochondrial transcription factor 1 (Tfam), cytochrome c oxidase subunit 1( CoxI), and mitochondrial cytochrome c oxidase subunit IV (CoxIV) [94]. Studies of energy metabolism in the GD mouse model demonstrated that ATP level and basal mitochondrial oxygen consumption were decreased in neuronal cells [30]. FD fibroblasts show the reduced activity of respiratory chain enzymes and decreased levels of cytochrome C oxidase activity [95]. In vivo studies of PBMC suggest that GD and FD display mitochondrial dysfunction due to malfunction of the mTOR pathway. Therefore, the deregulation of the autophagy–lysosomal pathway inhibits mTOR-mediated control of mitochondrial metabolism in GD and FD cells.

## 5. Conclusions

Altered sphingolipid metabolism complicates mitochondrial function in LSD, including GD and FD. Dysregulation of the autophagy–lysosomal signaling pathway plays an essential role in the inhibition of mitochondrial metabolism in GD and FD cells. In addition to autophagy–lysosomal abnormalities, a non-proportional ratio of sphingolipids (Cer, Gl-1, Gb-3) added a further layer of complexity to mitochondrial function due to defects of mitochondrial membrane composition. Therefore, the secondary functional alterations, as mitochondrial metabolism, play a vital role in GD and FD pathology, including neurological, cardiovascular, and renal complications.

## Figures and Tables

**Figure 1 jcm-09-01116-f001:**
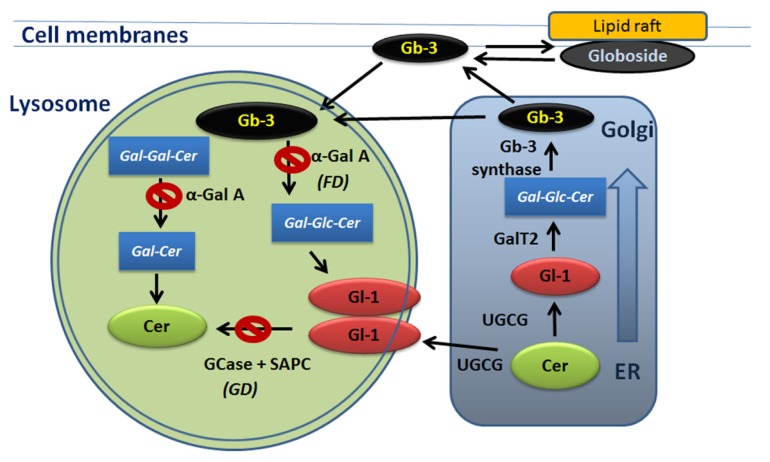
Sphingolipid metabolism in Gaucher and Fabry diseases (GD, FD). Ceramide (Cer), glucosylceramide (Gl-1) and globotriaosylceramide (Gb-3) shifts between endoplasmic reticulum ER, Golgi apparatus, lysosomes, and cellular membranes. UDP-Glucose Ceramide Glucosyltransferase (UGCG) converts Cer to Gl-1 in the ER. Gl-1 localized in the intralysosomal membrane is broken down by glucocerebrosidase (GCase) enzyme in the present of SAPC. Gb-3 is synthesized from lactosylceramide (Gal-Glc-Cer) by Golgi-localized enzyme Gb-3 synthase. Gal-Glc-Cer is synthesized by LacCer synthase (GalT2) from Gl-1. Lysosomal accumulation of Gl-1 is linked to GD. Lysosomal accumulation of Gb-3 is linked to FD.

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
