# Peer review of "Altered Sphingolipids Metabolism Damaged Mitochondrial Functions: Lessons Learned From Gaucher and Fabry Diseases"

_jcm, 2020, doi:10.3390/jcm9041116_

Round 1

Reviewer 1 Report

This review article "Altered sphingolipids metabolism damaged mitochondrial functions, lessons learned from Gaucher and Fabry diseases" highlights the interesting relationship between mitochondrial dysfunction and altered sphingolipid metabolism in the two most common lysosomal storage diseases, Fabry and Gaucher disease. This important topic once again demonstrates the failure of a multitude of cellular functions and processes due to a single genetic defect. What I feel is a little bit neglected is the importance of this particular pathophysiological process. It is certainly a bit speculative whether the described dysfunctions are of primary or secondary nature in the context oft he overall disease pathology, but it might be helpful to make an estimation based on a temporal approach. I would like to mention some further points of criticism, which should be worked on before the article is suitable for publication in JCM.

Main points:

-          The text contains insufficient literature references for numerous statements. Here some examples: Line 45 after „processes“; line 49 after „destabilization“; line Line 80 after „cargo“; line 81 after „induction“; line 143 after „pathways“; line 200 after „pathway“, and so on.

-          Many statements remain very superficial. At certain points the reader wonders what is written in the cited literature. As an example (lines 75-77): „Moreover, accumulation of SQSTM1/p62 in GD fibroblasts, neurons, or GBA knockout mouse models provides more evidence of abnormality in the autophagy-lysosomal process [7,17,24].“ What kind of abnormality is this? Does it go further than the statements in the previous sentence? Another example is (lines 99-101): „Therefore, instability of Cer/Gl-1 ratio in GD cells may play a role in the number and shape of lysosomes and may add a further layer of complexity to mitochondrial function [33].“ What phenotype could result from this „further layer“? The increased rigidity of membranes is mentioned, but what does it mean functionally?

-          Generally, the text should be carefully revised. As an example, this sentence is not comprehensive in its current form: „Neuronal accumulation of Gl-1 in mouse models of GD reveals accumulation of various autophagic cargo including dysfunctional mitochondria, ubiquitinated protein aggregates in autophagosomes and lysosomes in the brain, neurons, and astrocytes.“ Maybe „Neuronal phenotype involves dysfunctional mitochondria…based on Gl-1 accumulation“ or „Neuronal accumulation of Gl-1 in mouse models of GD leads to accumulation…“.

-         It was mentioned before that lysosomal pathway and mitophagy are entangled, but in lines 81-83 a new connection between proteasomal degradation and mitochondrial dysfunction is mentioned, but it is not explained in such a way that this bridge can be understood: „This effect may be attributed to the endoplasmic reticulum (ER) retention due to inability to correctly fold the mutant form of GBA protein and inhibition of proteasomal degradation [25].“

-          Ambroxol was also suggested as a therapy in Fabry disease. This should be mentioned and it can be used to bridge the two parts (lines 121-123).

-         Can the author speculate on the impact of less abundant „storage products and their role in the pathophysiology in the two diseases? This work mainly focusses on Gl-1 (Gaucher) and Gb3 / lyso-Gb3 (Fabry). But what about other neutral glycosphingolipids that are potential α–Gal A substrates, the dozens of Gb3 forms, etc.?

Minor point:

-        In lines 53-67 the general autophagy process is described. I recommend to use a different sub-header not referring to Gaucher disease here. I believe this paragraph is also important for the debate about Fabry. This should be restructured.

Author Response

Response to reviewers.

I thank you for this opportunity to respond to your comments and those of the two reviewers of manuscript  “Altered sphingolipids metabolism damaged mitochondrial functions, lessons learned from Gaucher and Fabry diseases ‘.

I thank the reviewers for their careful reading of manuscript and suggestions and questions for its improvement.

I have addressed each of the comments below in blue Arial 12 bold to distinguish responses form the Reviewers’ comments and questions. 

Reviewer 1.

This review article "Altered sphingolipids metabolism damaged mitochondrial functions, lessons learned from Gaucher and Fabry diseases" highlights the interesting relationship between mitochondrial dysfunction and altered sphingolipid metabolism in the two most common lysosomal storage diseases, Fabry and Gaucher disease.

This important topic once again demonstrates the failure of a multitude of cellular functions and processes due to a single genetic defect.

 What I feel is a little bit neglected is the importance of this particular pathophysiological process. It is certainly a bit speculative whether the described dysfunctions are of primary or secondary nature in the context oft he overall disease pathology, but it might be helpful to make an estimation based on a temporal approach.

 I would like to mention some further points of criticism, which should be worked on before the article is suitable for publication in JCM.

Main points:

-          The text contains insufficient literature references for numerous statements. Here some examples: Line 45 after „processes“; line 49 after „destabilization“; line Line 80 after „cargo“; line 81 after „induction“; line 143 after „pathways“; line 200 after „pathway“, and so on.

  • I have added references.

-          Many statements remain very superficial. At certain points the reader wonders what is written in the cited literature. As an example (lines 75-77): „Moreover, accumulation of SQSTM1/p62 in GD fibroblasts, neurons, or GBA knockout mouse models provides more evidence of abnormality in the autophagy-lysosomal process [7,17,24].“ What kind of abnormality is this? Does it go further than the statements in the previous sentence?

  • I apologize for the confusion. Yes, this statement is a continuation of the previous sentences, which stated that SQSTM1/p62 is an indicator of autophagy flux. The sentence was changed.

“Accumulation of SQSTM1/p62 is evidence of inhibition of autophagy-lysosomal process”.

  • Another example is (lines 99-101): „Therefore, instability of Cer/Gl-1 ratio in GD cells may play a role in the number and shape of lysosomes and may add a further layer of complexity to mitochondrial function [33].“ What phenotype could result from this „further layer“? The increased rigidity of membranes is mentioned, but what does it mean functionally?
  • Changed, the new text below:

“If instability of Cer/Gl-1 ratio in GD cells plays a role in the number and shape of lysosomes, then it may add a further layer of complexity to mitochondrial membrane structure too.  Moreover, Cer can self-assemble in the mitochondrial outer membrane to form large stable channels capable of releasing apoptotic proteins, such as cytochrome c [1,2]. Cer directly activates apoptosis via the formation of Cer channels in the mitochondrial outer membrane favoring BAK/BAX activation and regulates caspase 3 by compartmentalization in the late endosomes [3,4].

-          Generally, the text should be carefully revised. As an example, this sentence is not comprehensive in its current form: „Neuronal accumulation of Gl-1 in mouse models of GD reveals accumulation of various autophagic cargo including dysfunctional mitochondria, ubiquitinated protein aggregates in autophagosomes and lysosomes in the brain, neurons, and astrocytes.“ Maybe „Neuronal phenotype involves dysfunctional mitochondria…based on Gl-1 accumulation“ or „Neuronal accumulation of Gl-1 in mouse models of GD leads to accumulation…“.

  • The sentence has been changed. The text was revised.

- It was mentioned before that lysosomal pathway and mitophagy are entangled, but in lines 81-83 a new connection between proteasomal degradation and mitochondrial dysfunction is mentioned, but it is not explained in such a way that this bridge can be understood: „This effect may be attributed to the endoplasmic reticulum (ER) retention due to inability to correctly fold the mutant form of GBA protein and inhibition of proteasomal degradation [25].“

  • Explanation of how accumulation of unfolded mutated GBA is linked to activation of ER stress, UPR, and ERAD was added.

-   Ambroxol was also suggested as a therapy in Fabry disease. This should be mentioned and it can be used to bridge the two parts (lines 121-123).

  • The study from the Rolfs group was added [5].

-         Can the author speculate on the impact of less abundant „storage products and their role in the pathophysiology in the two diseases? This work mainly focusses on Gl-1 (Gaucher) and Gb3 / lyso-Gb3 (Fabry). But what about other neutral glycosphingolipids that are potential α–Gal A substrates, the dozens of Gb3 forms, etc.?

  • I agree that it is interesting to see the role of Gb-3 isoform, Gal-Gal-Cer in FD pathology. However, I can only guess what the role of Gb-3 analogs is in mitochondrial function. Lyso-Gb-3 analogs were identified as biomarkers for FD recently [6]. Up until today, there is a lot of speculation about the actual role of these molecules in FD pathology, including if we even need to add these molecules to the list of FD markers.   
  • Information about Gal-Gal-Cer Lyso-Gb-3 analogs was added to the text.

 Minor point:

-  In lines 53-67 the general autophagy process is described. I recommend to use a different sub-header not referring to Gaucher disease here. I believe this paragraph is also important for the debate about Fabry. This should be restructured.

  • The text was restructured. New sub-header: Mitophagy.
  1. Colombini, M. Ceramide channels and mitochondrial outer membrane permeability. Journal of bioenergetics and biomembranes 2017, 49, 57-64, doi:10.1007/s10863-016-9646-z.
  2. Siskind, L.J.; Kolesnick, R.N.; Colombini, M. Ceramide forms channels in mitochondrial outer membranes at physiologically relevant concentrations. Mitochondrion 2006, 6, 118-125, doi:10.1016/j.mito.2006.03.002.
  3. Chipuk, J.E.; McStay, G.P.; Bharti, A.; Kuwana, T.; Clarke, C.J.; Siskind, L.J.; Obeid, L.M.; Green, D.R. Sphingolipid metabolism cooperates with BAK and BAX to promote the mitochondrial pathway of apoptosis. Cell 2012, 148, 988-1000, doi:10.1016/j.cell.2012.01.038.
  4. Jain, A.; Beutel, O.; Ebell, K.; Korneev, S.; Holthuis, J.C. Diverting CERT-mediated ceramide transport to mitochondria triggers Bax-dependent apoptosis. Journal of cell science 2017, 130, 360-371, doi:10.1242/jcs.194191.
  5. Lukas, J.; Pockrandt, A.M.; Seemann, S.; Sharif, M.; Runge, F.; Pohlers, S.; Zheng, C.; Glaser, A.; Beller, M.; Rolfs, A., et al. Enzyme enhancers for the treatment of Fabry and Pompe disease. Molecular therapy : the journal of the American Society of Gene Therapy 2015, 23, 456-464, doi:10.1038/mt.2014.224.
  6. Boutin, M.; Menkovic, I.; Martineau, T.; Vaillancourt-Lavigueur, V.; Toupin, A.; Auray-Blais, C. Separation and Analysis of Lactosylceramide, Galabiosylceramide, and Globotriaosylceramide by LC-MS/MS in Urine of Fabry Disease Patients. Analytical chemistry 2017, 89, 13382-13390, doi:10.1021/acs.analchem.7b03609.
  7. TakashiKandacMasaakiUematsucYoshihikoIkedadHatsueIshibashi-UedadChikaoYutania, T.N.S.-i.N.S. Myocardial fibrosis pathology in Anderson–Fabry disease: Evaluation of autopsy cases in the long- and short-term enzyme replacement therapy, and non-therapy case. IJC Metabolic & Endocrine

 2016, Volume 12, Pages 46-51.

Reviewer 2 Report

The manuscript by Margarita Ivanova describes the pathologies of Gaucher and Fabry diseases, and further addresses how mitochondrial functions are impacted in these lysosomal storage diseases.

The text is well written, and the pathology description is accurate. In regard to the mitochondrial aspects, the manuscript falls a bit short, and could use some revamping.

  • several recent studies and reviews addressed the effect of lysosomal malfunction on mitochondrial biogenesis and function, and this review, although focused in GD and FD, would benefit from integrating the knowledge obtained with other lysosomal diseases, some of them also from sphingolipid catabolism
  • line 54 - the consequences for mitochondria are not just due to impaired mitophagy (see my previous note)
  • line 69 - fusion of mitochondria and lysosomes is not something that happens biologically, so this should be rephrased
  • line 71 - decreased LC3II is an indicator of decreased autophagosome formation OR excessive autophagosome degradation. Per se, cannot be used as a determinant of autophagic flux
  • lines 100-103 - it would be nice to discuss briefly how and why the number and shape of lysosomes may impact mitochondrial function, and how the ceramide in the lysosome may end up in mitochondria
  • line 154 - decreased total area of mitochondria is suggestive of decreased mitochondrial mass. Is that the case? It should be described with a bit more detail.
  • line 158 - the loss od SOD2 activity does not increase ROS production, but increases the steady-state ROS levels
  • line 176 - effects of impaired sphingolipid metabolism on mitochondria have been explored in mechanistic detail, it would be beneficial to the interest of this review to compare the findings in kidney with findings observed in other tissues of sphingolipidosis such as liver and brain
  • line 190 - mTORC is not a sensor of energy status - that is the role of AMPK. mTORC is more a sensor of nutrients and growth factors that also receives an input from AMPK. This should be corrected.
  • line 191 - AMPK does not respond only to AMP/ATP, but also to different stresses such as Ca2+ spikes, ROS, and some metabolites
  • lines 192-193 - when mTORC1 phosphorylates TFEB, it gets excluded from the nucleus, so it does not trigger autophagy. Activation of TFEB and autophagy happen when mTORC1 is down-regulated.

Author Response

Response to reviewers.

I thank you for this opportunity to respond to your comments and those of the two reviewers of manuscript  “Altered sphingolipids metabolism damaged mitochondrial functions, lessons learned from Gaucher and Fabry diseases ‘.  . 

I thank the reviewers for their careful reading of manuscript and suggestions and questions for its improvement.

I have addressed each of the comments below in blue Arial 12 bold to distinguish responses form the Reviewers’ comments and questions. 

Reviewer 2.

Comments and Suggestions for Authors

The manuscript by Margarita Ivanova describes the pathologies of Gaucher and Fabry diseases and further addresses how mitochondrial functions are impacted in these lysosomal storage diseases.

The text is well written, and the pathology description is accurate. In regard to the mitochondrial aspects, the manuscript falls a bit short, and could use some revamping.

  • several recent studies and reviews addressed the effect of lysosomal malfunction on mitochondrial biogenesis and function, and this review, although focused in GD and FD, would benefit from integrating the knowledge obtained with other lysosomal diseases, some of them also from sphingolipid catabolism
  • There are approximately 50 metabolic disorders with defected lysosomal functions, eleven of which are linked to sphingolipid metabolism. In this review, I focused on the most common disorders, Gaucher and Fabry, because these are the two most common sphingolipid disorders and are prevalent in adults, not just children. This is also my area of expertise, and so I don’t feel it is my place to discuss other lysosomal diseases I am not as knowledgeable about.
  • line 54 - the consequences for mitochondria are not just due to impaired mitophagy (see my previous note)
  • The sentence was re-phrased.
  • line 69 - fusion of mitochondria and lysosomes is not something that happens biologically, so this should be rephrased
  • The sentence was changed.
  • line 71 - decreased LC3II is an indicator of decreased autophagosome formation OR excessive autophagosome degradation. Per se, cannot be used as a determinant of autophagic flux
  • The sentence was changed.
  • lines 100-103 - it would be nice to discuss briefly how and why the number and shape of lysosomes may impact mitochondrial function, and how the ceramide in the lysosome may end up in mitochondria
  • Sorry for the misinterpretation. Accumulation of Gl-1 in cells has a significant effect on all membrane structures, not just cytoplasmic and lysosomal membrane.
  • Changed, below the new text:

“If instability of Cer/Gl-1 ratio in GD cells plays a role in the number and shape of lysosomes, then it may add a further layer of complexity to mitochondrial membrane structure too.  Moreover, Cer can self-assemble in the mitochondrial outer membrane to form large stable channels capable of releasing apoptotic proteins, such as cytochrome c [1,2]. Cer directly activates apoptosis via the formation of Cer channels in the mitochondrial outer membrane favoring BAK/BAX activation and regulates caspase 3 by compartmentalization in the late endosomes [3,4].

  • line 154 - decreased total area of mitochondria is suggestive of decreased mitochondrial mass. Is that the case? It should be described with a bit more detail.
  • This data is from autopsy reports. More information included:
  • “Electron microscopic evaluation of cardiomyocytes from FD patient's hearts demonstrated that the percentage area of mitochondria in the cytoplasm was reduced and Gb-3 accumulation was increased in 2 out of 3 cases [7]
  • line 158 - the loss of SOD2 activity does not increase ROS production, but increases the steady-state ROS levels
  • The sentence was changed.
  • line 176 - effects of impaired sphingolipid metabolism on mitochondria have been explored in mechanistic detail, it would be beneficial to the interest of this review to compare the findings in kidney with findings observed in other tissues of sphingolipidosis such as liver and brain
  • Cardiac disease and/or renal failure are lead to early death of Fabry patients. Because of this, majority of investigators study cellular aspects of disease using heart or kidney models. I did not find publications about mitochondria function in liver or brain cells related to Fabry disease. However, I included the cellular study of Gb-3 accumulation in medullary thick ascending limbs. Line 233-241.
  • line 190 - mTORC is not a sensor of energy status - that is the role of AMPK. mTORC is more a sensor of nutrients and growth factors that also receives an input from AMPK. This should be corrected.
  • The sentence was changed
  • line 191 - AMPK does not respond only to AMP/ATP, but also to different stresses such as Ca2+ spikes, ROS, and some metabolites
  • This text was focusing on the mechanism of mTOR inhibition, not AMP/ATP.
  • lines 192-193 - when mTORC1 phosphorylates TFEB, it gets excluded from the nucleus, so it does not trigger autophagy. Activation of TFEB and autophagy happen when mTORC1 is down-regulated.
  • The sentence was changed
  1. Colombini, M. Ceramide channels and mitochondrial outer membrane permeability. Journal of bioenergetics and biomembranes 2017, 49, 57-64, doi:10.1007/s10863-016-9646-z.
  2. Siskind, L.J.; Kolesnick, R.N.; Colombini, M. Ceramide forms channels in mitochondrial outer membranes at physiologically relevant concentrations. Mitochondrion 2006, 6, 118-125, doi:10.1016/j.mito.2006.03.002.
  3. Chipuk, J.E.; McStay, G.P.; Bharti, A.; Kuwana, T.; Clarke, C.J.; Siskind, L.J.; Obeid, L.M.; Green, D.R. Sphingolipid metabolism cooperates with BAK and BAX to promote the mitochondrial pathway of apoptosis. Cell 2012, 148, 988-1000, doi:10.1016/j.cell.2012.01.038.
  4. Jain, A.; Beutel, O.; Ebell, K.; Korneev, S.; Holthuis, J.C. Diverting CERT-mediated ceramide transport to mitochondria triggers Bax-dependent apoptosis. Journal of cell science 2017, 130, 360-371, doi:10.1242/jcs.194191.
  5. Lukas, J.; Pockrandt, A.M.; Seemann, S.; Sharif, M.; Runge, F.; Pohlers, S.; Zheng, C.; Glaser, A.; Beller, M.; Rolfs, A., et al. Enzyme enhancers for the treatment of Fabry and Pompe disease. Molecular therapy : the journal of the American Society of Gene Therapy 2015, 23, 456-464, doi:10.1038/mt.2014.224.
  6. Boutin, M.; Menkovic, I.; Martineau, T.; Vaillancourt-Lavigueur, V.; Toupin, A.; Auray-Blais, C. Separation and Analysis of Lactosylceramide, Galabiosylceramide, and Globotriaosylceramide by LC-MS/MS in Urine of Fabry Disease Patients. Analytical chemistry 2017, 89, 13382-13390, doi:10.1021/acs.analchem.7b03609.
  7. TakashiKandacMasaakiUematsucYoshihikoIkedadHatsueIshibashi-UedadChikaoYutania, T.N.S.-i.N.S. Myocardial fibrosis pathology in Anderson–Fabry disease: Evaluation of autopsy cases in the long- and short-term enzyme replacement therapy, and non-therapy case. IJC Metabolic & Endocrine

 2016, Volume 12, Pages 46-51.

Round 2

Reviewer 1 Report

Thanks for revising the article. In general my points have been adequately edited. However, I have found some minor points that should be addressed.

(1)    Page 3, line 90-f.:“Impaired mitochondrial clearance due to inhibition of autophagosome-lysosome fusion was described in neurons and fibroblasts of GBAL444P/wt mice, and brain tissue from Parkinson’s patients with GBA mutations” It should be noted that this is not Gaucher disease. Heterozygotes do not develop GD. Thus, this is not a finding related to sphingolipid accumulation. This could be misinterpreted by an impartial reader.

(2)    Page 3, line 100: “Recently emerging studies demonstrated that mechanism inhibition of mitophagy in GD is dual.“ I do not necessarily see the causality between protein misfolding / ERAD and mitophagy inhibition. The theorem implies a causality that the phenomena observed in GD "Degradation block of autophagic cargo" and GCase protein accumulation in ER" result in mitophagy inhibition. Here, (possibly) independent processes are mixed together. Definitely, protein misfolding is at most secondarily associated with sphingolipid accumulation. Thus, Sphingolipids action is not involved, but is the topic of the article.

(3)    Typo: leasd (line 108)

(4)    Typo: consuption (line 118)

(5)    Page 4, lines 156-58: „Additionally, Rolfs team demonstrated that ambroxol is effective in enhancing activity of mutant enzyme in Fabry disease, suggesting that ambroxol could potentially be used in the treatment of different LSDs.“ „Rolfs team“ sounds rather non-Standard. Also the McNeill article claims that ABX may potentially be used in other LSDs, because the CLEAR network genes were differentially upregulated by ABX treatment. The Rolfs paper is just further experimental evidence that their assumption may be right.

(6)    Typo: revialed (line 174)

(7)    Grammar: „However(, the) role of Lyso-Gb-3 analogues in FD cellular pathology (is) unknown.“ (line 176). There are several examples, but I am generally not qualified to revise this. However, this sentence stands as an example for omitted articles and auxiliary verbs, so that some sentences read like rudiments. Please have an English native editor edit it again.

(8)    „Strictly“ (line 191). Better „systematically“.

(9)    induces à induced (line 208)

Author Response

Reviewer 1.

Thanks for revising the article. In general my points have been adequately edited. However, I have found some minor points that should be addressed.

 (1)    Page 3, line 90-f.:“Impaired mitochondrial clearance due to inhibition of autophagosome-lysosome fusion was described in neurons and fibroblasts of GBAL444P/wt mice, and brain tissue from Parkinson’s patients with GBA mutations” It should be noted that this is not Gaucher disease. Heterozygotes do not develop GD. Thus, this is not a finding related to sphingolipid accumulation. This could be misinterpreted by an impartial reader.

I agree with the reviewer that Parkinson’s studies could make a kind of confusion. I removed this sentence, despite the fact that a lot of investigations about mitochondria function in Parkinson’s patients with GBA heterozygous mutations.

The old text: Impaired mitochondrial clearance due to inhibition of autophagosome-lysosome fusion was described in neurons and fibroblasts of GBAL444P/WT mice, and brain tissue from Parkinson’s patients with GBA mutations [1]. 

New text: Lysosomal dysfunction in LSD is associated with mitochondrial dysregulation and the accumulation of damaged mitochondria. What happens with the mitophagy process in GD when lysosomes do not function properly?  Increased mitochondrial fragmentation due to inhibition of autophagy was described in midbrain neurons and astrocytes in GBA-/- mice  [2].

(2)    Page 3, line 100: “Recently emerging studies demonstrated that mechanism inhibition of mitophagy in GD is dual.“ I do not necessarily see the causality between protein misfolding / ERAD and mitophagy inhibition. The theorem implies a causality that the phenomena observed in GD "Degradation block of autophagic cargo" and GCase protein accumulation in ER" result in mitophagy inhibition. Here, (possibly) independent processes are mixed together. Definitely, protein misfolding is at most secondarily associated with sphingolipid accumulation. Thus, Sphingolipids action is not involved, but is the topic of the article.

This point of view is still being debated in the research community.  I would like to include and touch on different opinions in this review even though it’s not everyone agrees. However myself, I agree with researchers that the molecular mechanism of inhibition of mitophagy and autophagy in GD is dual. Reason for this that sphingolipids metabolism altered in GD not only due to Gl-1 accumulation in lysosomes but also the buildup of a mutated enzyme in the cytoplasm, which clog the ERAD.  

 (3)    Typo: leasd (line 108)

New: Moreover, accumulation of unfolded mutated GBA protein leads to inhibition of alpha-synuclein degradation, which  directs alpha-synuclein aggregation in GD cells.

(4)    Typo: consuption (line 118)

New. The loss of mitochondrial function, inhibition of oxygen consumption due to decreasing ѰM has been demonstrated in GD neuroblastoma cells [35], fibroblasts from GD patients (L444P/L444P

(5)    Page 4, lines 156-58: „Additionally, Rolfs team demonstrated that ambroxol is effective in enhancing activity of mutant enzyme in Fabry disease, suggesting that ambroxol could potentially be used in the treatment of different LSDs.“ „Rolfs team“ sounds rather non-Standard. Also the McNeill article claims that ABX may potentially be used in other LSDs, because the CLEAR network genes were differentially upregulated by ABX treatment. The Rolfs paper is just further experimental evidence that their assumption may be right.

Correct, McNeill study claimed that ABX could be used for treatment not only GD but also Parkinson’s patients due to GCase chaperone activity, increasing expression of TFEB, which is associated with activation of components of the CLEAR network. Rolf group demonstrated that ABX increased not only GCase enzyme activity but also enhanced α-Gal-A and GAA enzyme activities (mean potentially, Fabry and Pompe patients could be treated with ambroxol).

McNeill article was included in the text. “At pH 4.3, near lysosomal pH, ambroxol does not inhibit the enzyme but actually becomes an activator of GCase activity [3,4].”

The sentence was changed: before: “Additionally, the Rolfs team showed that ambroxol is effective in enhancing activity of mutant enzyme in Fabry disease, suggesting that ambroxol could potentially be used in the treatment of different LSDs [57].”

New sentence: Additionally, ambroxol showed significant enhancing activity of wild-type and mutant (p.A156V and p.R301Q) forms of α-Gal A enzyme, suggesting that ambroxol could potentially be used in the treatment of Fabry diseases [57].

(6)    Typo: revialed (line 174)

New: Recent metabolic studies showed that Lyso-Gb-3 analogs (Lyso-Gb-3(-28),

(7)    Grammar: „However(, the) role of Lyso-Gb-3 analogues in FD cellular pathology (is) unknown.“ (line 176). There are several examples, but I am generally not qualified to revise this. However, this sentence stands as an example for omitted articles and auxiliary verbs, so that some sentences read like rudiments. Please have an English native editor edit it again.

New sentence: Recent metabolic studies showed that Lyso-Gb-3 analogs (Lyso-Gb-3(-28), Lyso-Gb-3(-2), Lyso-Gb-3(+16), Lyso-Gb-3(+18), Lyso-Gb-3(+34), Lyso-Gb-3(+50) higher in GLA knockout mice and also present in plasma and urine samples in FD patients with highest Lyso-Gb-3 levels [64-66]. However role of Lyso-Gb-3 analogs in FD cellular pathology unknown. 

(8)    „Strictly“ (line 191). Better „systematically“.

New sentence However, unfortunately, mitochondrial function and energy metabolism were never systematically studied in aspects of FD pathology.

(9)    induces à induced (line 208)

New sentence Gb-3 accumulation induced ROS production, suppressed mitochondrial antioxidant SOD2, enhanced AMPK activation in vascular endothelial cells and iPS cells derived from FD patients

  1. Li, H.; Ham, A.; Ma, T.C.; Kuo, S.H.; Kanter, E.; Kim, D.; Ko, H.S.; Quan, Y.; Sardi, S.P.; Li, A., et al. Mitochondrial dysfunction and mitophagy defect triggered by heterozygous GBA mutations. Autophagy 2019, 15, 113-130, doi:10.1080/15548627.2018.1509818.
  2. Osellame, L.D.; Duchen, M.R. Defective quality control mechanisms and accumulation of damaged mitochondria link Gaucher and Parkinson diseases. Autophagy 2013, 9, 1633-1635, doi:10.4161/auto.25878.
  3. Maegawa, G.H.; Tropak, M.B.; Buttner, J.D.; Rigat, B.A.; Fuller, M.; Pandit, D.; Tang, L.; Kornhaber, G.J.; Hamuro, Y.; Clarke, J.T., et al. Identification and characterization of ambroxol as an enzyme enhancement agent for Gaucher disease. The Journal of biological chemistry 2009, 284, 23502-23516, doi:10.1074/jbc.M109.012393.
  4. McNeill, A.; Magalhaes, J.; Shen, C.; Chau, K.Y.; Hughes, D.; Mehta, A.; Foltynie, T.; Cooper, J.M.; Abramov, A.Y.; Gegg, M., et al. Ambroxol improves lysosomal biochemistry in glucocerebrosidase mutation-linked Parkinson disease cells. Brain : a journal of neurology 2014, 137, 1481-1495, doi:10.1093/brain/awu020.
